# Effects of Darbepoetin Alfa and Ferric Derisomaltose Plus Darbepoetin Alfa in Functional Iron-Deficiency Anemia

**DOI:** 10.3390/ijms26052203

**Published:** 2025-02-28

**Authors:** Sung-Hwa Sohn, Heejung Sul, Bumjun Kim, Daeyoung Zang

**Affiliations:** 1Hallym Translational Research Institute, College of Medicine, Hallym University, Anyang-si 14068, Gyeonggi-do, Republic of Korea; iisupy@korea.ac.kr (S.-H.S.); glwjd82@naver.com (H.S.); 2Department of Internal Medicine, Hallym University Medical Center, College of Medicine, Hallym University, Anyang-si 14068, Gyeonggi-do, Republic of Korea; getwisdom1025@gmail.com

**Keywords:** darbepoetin alfa, ferric derisomaltose, chemotherapy induced functional iron-deficiency anemia

## Abstract

Functional iron-deficiency anemia (FIDA) is a side effect of many cancer treatments, occurring when chemotherapy drugs damage bone marrow cells, which are responsible for producing red blood cells, due to the myelosuppressive effects of chemotherapy, or to the cancer itself. This study was performed to compare the effects of darbepoetin alfa alone, or in combination with ferric derisomaltose in cancer patients with FIDA, and to elucidate the mechanism underlying the effects in F36E cells. F36E cells treated with darbepoetin alfa showed increased cell viability. AML and GC cells treated with darbepoetin alfa, ferric derisomaltose, or ferric derisomaltose plus darbepoetin alfa showed no induction of apoptosis. The effects of these drugs on the anticancer efficacy of PTX chemotherapy were examined by analyzing cell viability and induction of apoptosis. Darbepoetin alfa, ferric derisomaltose, and ferric derisomaltose plus darbepoetin alfa showed no significant inhibitory effects on the apoptosis-inducing activity of PTX in GC cell lines. Patients with chemotherapy-induced FIDA in Group I receiving ferric derisomaltose plus darbepoetin alfa showed higher hemoglobin levels, transferrin saturation, and ferritin levels compared to those in Group II, treated with darbepoetin alfa alone. In cancer patients with FIDA, the prognosis of anemia treatment was better in the ferric derisomaltose plus darbepoetin alfa combination group than in the group receiving darbepoetin alfa monotherapy.

## 1. Introduction

Anemia is common in patients with cancer receiving chemotherapy and is known to cause not only fatigue and adversely affect quality of life but also to reduce treatment response rates and even overall survival rates [1,2,3]. Functional iron-deficiency anemia (FIDA) is a condition in which the body has sufficient iron stores but cannot use it effectively, which is distinct from iron-deficiency anemia where the level of iron in the body is low. The goal of treating chemotherapy-induced anemia (CIA) is to alleviate anemia-related symptoms and improve quality of life by correcting the anemia.

The National Comprehensive Cancer Network (NCCN) and European Society for Medical Oncology (ESMO) guidelines recommend the transfusion of red blood cells (RBCs), administration of erythropoiesis-stimulating agents (ESAs), and use of iron supplements to treat CIA, which can increase RBC production in bone marrow (BM) by activating the erythropoietin receptor (EpoR) on erythrocytic progenitor cells [4,5,6,7]. However, each option has its limitations, and novel approaches are required.

A recently developed third-generation high-dose intravenous iron preparation was shown to be more convenient and to have reduced side effects, including the risk of anaphylaxis, by allowing the administration of a large dose of iron in a single infusion. Its efficacy in treating iron-deficiency anemia in general patients has been confirmed, and it is now in active use in clinical practice [8]. Combination therapy with ESAs and intravenous iron is recommended for the treatment of patients with CIA, with functional iron deficiency. The iron preparations used in previous studies demonstrating the efficacy of combination therapy were second-generation general intravenous iron preparations, such as ferric gluconate, iron sucrose, and iron dextran. Third-generation high-dose intravenous iron preparations significantly improve the convenience of treatment, with fewer side effects compared to second-generation preparations, and are advantageous for the treatment of anemia in patients with advanced cancer who have difficulties in repeated visits to medical institutions due to the deterioration of their general condition [9,10]. Darbepoetin alfa binds to the EpoR on erythroid progenitor cells and stimulates increasing RBC production and RBC hemoglobinization (erythropoiesis) by the same mechanism as endogenous erythropoietin [11,12]. Ferric derisomaltose (third-generation high-dose intravenous iron supplement) releases iron and is then transported to erythroid precursor cells for incorporation into the hemoglobin [13,14]. However, there have been few studies on third-generation high-dose intravenous iron supplements, and no studies on their mechanisms of action and efficacy when administered alone or in combination with ESAs.

This study was performed to compare the therapeutic mechanism of action and effects of combination therapy with ferric derisomaltose and darbepoetin alfa, an ESA, with ESA monotherapy in erythroid cell lines. In addition, we examined whether the combination of ferric derisomaltose plus darbepoetin alfa or darbepoetin alfa monotherapy adversely affected the anticancer efficacy of PTX chemotherapy in gastric cancer (GC) cell lines.

## 2. Results

### 2.1. Effects of Darbepoetin Alfa and Ferric Derisomaltose on Acute Myeloid and Erythroid Leukemia Cell Lines

We investigated the effects of darbepoetin alfa and ferric derisomaltose on the proliferation of myeloid and erythroid cell lines. The dose dependency of the effects of ferric derisomaltose and darbepoetin alfa were examined in K562 and F36E cells (Figure 1). In these in vitro models, cell viability increased only in F36E erythroid cells treated with darbepoetin alfa.

### 2.2. Effects of Darbepoetin Alfa and Ferric Derisomaltose on Gastric Cancer Cell Lines

We investigated whether darbepoetin alfa and ferric derisomaltose affect the proliferation of cancer cell lines. The dose dependency of the effects of darbepoetin alfa and ferric derisomaltose were examined in the GC cell lines SNU620 and SNU638 cells (Figure 2). Darbepoetin alfa and ferric derisomaltose did not affect the cell viability of these GC cell lines in vitro.

### 2.3. Effects of Ferric Derisomaltose and Darbepoetin Alfa on PTX-Induced Apoptosis of Erythroid Cells

We next examined the effects of ferric derisomaltose and darbepoetin alfa on the anticancer agent PTX-induced apoptosis of F36E cells. Treatment with PTX decreased the cell viability of F36E erythroid cells in a dose-dependent manner with a half-maximal inhibitory concentration (IC50) of 5.61 nM (Figure 3A).

After the viability of the erythroid cell line was reduced by 80% with 10 nM PTX, viability was examined after treatment with ferric derisomaltose or darbepoetin alfa in serial dilutions from 0.000001 to 100 ng/mL (Figure 3B). Treatment with darbepoetin alfa increased cell viability of F36E erythroid cells in a dose-dependent manner. In particular, darbepoetin alfa at concentrations of 10 ng/mL or higher resulted in more than 100% increases in the viability of F36E erythroid cells treated with 10 nM PTX. Therefore, a concentration of 10 ng/mL was selected for darbepoetin alfa.

Viability was examined after treating F36E erythroid cells with 10 nM of PTX, 10 ng/mL of darbepoetin alfa, and serial dilutions of ferric derisomaltose from 0.000001 to 100 ng/mL (Figure 3C). An increase in viability of approximately 20% was observed in cultures when a high concentration of 100 ng/mL ferric derisomaltose was added to 10 nM PTX plus 10 ng/mL darbepoetin alfa. Therefore, a concentration of 100 ng/mL was selected for ferric derisomaltose.

### 2.4. Effects of Ferric Derisomaltose and Darbepoetin Alfa on the Efficacy of PTX in Gastric Cancer Cell Lines

The effects of ferric derisomaltose and darbepoetin alfa on the anticancer efficacy of PTX were examined in the GC cell lines SNU620 and SNU638 (Figure 4). Treatment with PTX decreased cell viability in the two GC cells in a dose-dependent manner with IC50 values of 5.61 and 14.23 nM for SNU620 and SNU638 cells, respectively.

After reducing the viability of the GC cell line by ~40–50% with 10 nM of PTX, the viability was examined after treatment with ferric derisomaltose or darbepoetin alfa in serial dilutions from 1 × 10^−6^ to 100 ng/mL (Figure 4). The results showed that treatment with ferric derisomaltose or darbepoetin alfa did not inhibit PTX activity (i.e., reduction of cell viability).

### 2.5. Effects of Ferric Derisomaltose and Darbepoetin Alfa on Apoptosis of Myeloid and Erythroid Cell Lines

We examined whether ferric derisomaltose and darbepoetin alfa induced apoptosis in myeloid K562 and erythroid F36E cells (Figure 5 and Appendix A). The results confirmed that neither ferric derisomaltose nor darbepoetin alfa induced apoptosis or necrosis in either of these cell lines.

### 2.6. Effects of Ferric Derisomaltose and Darbepoetin Alfa on PTX-Induced Apoptosis in GC Cell Lines

The effects of ferric derisomaltose and darbepoetin alfa on PTX-induced apoptotic cell death in the GC cell lines SNU620 and SNU638 were examined. Briefly, 10 nM PTX induced cell death (apoptosis and necrosis) in cultures of SNU620 and SNU638 cells (Figure 6 and Appendix A). Both SNU620 and SNU638 cells showed high cell death rates following treatment with PTX, PTX plus ferric derisomaltose, PTX plus darbepoetin alfa, and PTX plus ferric derisomaltose plus darbepoetin alfa. Darbepoetin alfa, ferric derisomaltose, and ferric derisomaltose plus darbepoetin alfa did not significantly inhibit the apoptosis-inducing effect of PTX. We also confirmed that ferric derisomaltose and darbepoetin alfa did not induce apoptosis in these two GC cell lines.

### 2.7. Effects of Darbepoetin Alfa vs. Ferric Derisomaltose Plus Darbepoetin Alfa on F36E Erythroid Cells

Chemotherapy can decrease the production of EPO, a humoral regulator of erythropoiesis produced primarily in the kidneys, which acts to signal RBC production in the bone marrow. Such a decrease in the level of EPO can lead to the worsening of anemia in patients undergoing chemotherapy [15]. We measured EPO protein levels in the culture medium after 48 h of treatment of F36E cells with ferric derisomaltose, darbepoetin alfa, and ferric derisomaltose plus darbepoetin alfa. F36E cells treated with darbepoetin alfa or ferric derisomaltose plus darbepoetin alfa showed increased EPO protein production, with combination treatment leading to the highest EPO levels in the medium (Figure 7A). Changes in gene expression in erythroid cell lines treated with darbepoetin alfa vs. ferric derisomaltose plus darbepoetin alfa were examined by RNA-seq analysis (Figure 7B,C). When ferric derisomaltose was added to F36E erythroid cell lines treated with darbepoetin alfa, increases were detected in the expression of genes involved in myeloid cell development, transmembrane transport, and protein processing (Figure 7B), whereas the expression levels of genes involved in endothelial cell morphogenesis, endothelial cell chemotaxis, and the transmembrane receptor protein serine/threonine kinase-signaling pathway decreased (Figure 7C).

### 2.8. Effects of Ferric Derisomaltose Plus Darbepoetin Alfa and Darbepoetin Alfa Monotherapy in Patients with Chemotherapy-Induced Functional Iron-Deficiency Anemia

A total of 27 cancer patients with FIDA were enrolled in this study, consisting of 14 patients treated with ferric derisomaltose plus darbepoetin alfa (Group I) and 13 patients treated with darbepoetin alfa only (Group II). After a 12-week follow-up period, Group I showed higher hemoglobin levels, transferrin saturation, and ferritin levels than Group II (Figure 8). However, total iron-binding capacity (TIBC) was slightly higher in Group II. These observations indicated greater relief of anemia in Group I than in Group II.

## 3. Discussion

CIA is one of the many side effects of chemotherapy drugs, mainly caused by the inhibition of hematopoiesis in the bone marrow or reduced lifespan of RBCs [2,3]. Patients often have difficulty digesting food during chemotherapy, which can lead to a deficiency of nutrients important for erythropoiesis, such as iron, vitamin B12, and folic acid. Iron deficiency is closely related to anemia. Iron-deficiency anemia (IDA) includes absolute iron-deficiency anemia (AIDA), defined as a decrease in the body’s iron stores, and FIDA, where the body has sufficient iron stored but cannot use it effectively [16]. Chemotherapy also reduces EPO production, leading to a worsening of anemia [15]. This study was performed to identify effective treatment regimens and determine the mechanism of action of such treatment in patients with chemotherapy-induced FIDA.

Treatment options for IDA include EPO injections, iron supplements, and RBC transfusions. In patients with CIA who have functional iron deficiency, it is recommended to consider adding iron supplementation to the ESA treatment. Many prospective studies have indicated that the combination of ESA with iron supplementation shows greater efficacy than ESA monotherapy [17,18,19]. Second-generation general intravenous iron preparations, such as ferric gluconate, iron sucrose, and iron dextran have been shown to be effective and are recommended for use in combination with ESAs and intravenous iron, but these are accompanied by side effects and require frequent administration during treatment. In this study, we found that the third-generation high-dose intravenous iron preparation, ferric derisomaltose, which had fewer side effects and did not require repeated visits to medical institutions due to the high doses compared to second-generation intravenous iron preparations [9,10], increased hemoglobin level, transferrin saturation, and ferritin level after combined treatment with the ESA darbepoetin alfa in patients with CIA (Figure 8). To elucidate the mechanism underlying its efficacy, we performed RNA-seq analysis in the EPO-dependent human acute erythroid leukemia cell line F36E cells. EPO is required for the activation of the JAK2/STAT5 pathway, which activates genes fundamental for erythroid progenitor proliferation, differentiation, and survival [20,21]. In this study, F36E cells treated with ferric derisomaltose plus darbepoetin alfa showed greater increases in EPO protein and gene expression than cells treated with darbepoetin alfa alone (Figure 7). Treatment of F36E cells with the chemotherapy agent PTX decreased cell viability by about 80%. Treatment of these PTX-treated cells with ferric derisomaltose plus darbepoetin alfa showed an increase in cell viability of more than 100% in comparison to darbepoetin alfa alone (Figure 3). Darbepoetin alfa and ferric derisomaltose did not induce the apoptosis of myeloid K562 cells, erythroid F36E cells, or the GC cell lines SNU620 and SNU638 (Figure 5 and Figure 6). In addition, darbepoetin alfa and ferric derisomaltose did not inhibit the induction of apoptosis or increase the cell viability of PTX-treated GC cell lines (Figure 4 and Figure 6).

Compared with darbepoetin alfa monotherapy, ferric derisomaltose plus darbepoetin alfa showed a greater increase in the growth of EPO-dependent F36E erythroid cells. Ferric derisomaltose plus darbepoetin alfa combination treatment was associated with a higher level of erythropoiesis than darbepoetin alfa monotherapy.

## 4. Materials and Methods

### 4.1. Drug Preparation

The third-generation high-dose intravenous iron supplement ferric derisomaltose (Monofer; Pharmbio Korea, Seoul, Republic of Korea), the ESA darbepoetin alfa (CKD-11101, Nesbell; Chong Kun Dang Pharm, Seoul, Republic of Korea) [22], and the microtubule depolymerization inhibitor paclitaxel (PTX) (Selleck Chemicals, Houston, TX, USA) were obtained from the sources shown.

### 4.2. Cell Lines and Cell Culture

The human GC cell lines SNU620 and SNU638 and human acute myeloid leukemia (AML) without maturation (M1) [23] cell line K562 were obtained from the Korean Cell Line Bank (KCLB, Seoul, Republic of Korea). K562 cells were maintained in RPMI 1640 medium supplemented with 10% fetal bovine serum (FBS) and 1% penicillin/streptomycin. The erythropoietin (EPO)-dependent human myelodysplastic syndrome (acute erythroid leukemia, M6) cell line F36E was obtained from the Japanese Cancer Research Resources Bank (Tokyo, Japan). F36E cells were maintained in RPMI 1640 medium supplemented with 5% FBS and 1% penicillin/streptomycin to which 5 IU/mL darbepoetin alfa was added (BD Biosciences, Palo Alto, CA, USA). Cell culture was performed using standard procedures.

### 4.3. Cell Viability Assays

Cell viability of SNU620, SNU638, K562, and F36E cells was examined by MTS tetrazolium assay for cells treated with ferric derisomaltose and darbepoetin alfa at concentrations of 100, 10, 1, 0.1, 1 × 10^−2^, 1 × 10^−3^, 1 × 10^−4^, 1 × 10^−5^, and 1 × 10^−6^ ng/mL, and PTX at concentrations of 100, 10, 1, 0.1, 1 × 10^−2^, 1 × 10^−3^, 1 × 10^−4^, 1 × 10^−5^, and 1 × 10^−6^ µM for 48 h. MTS assay was performed using a CellTiter 96^®^AQueous Non-Radioactive Cell Proliferation Assay kit (Promega, Madison, WI, USA).

### 4.4. Apoptosis Analysis

K562 and F36E cells were seeded in 6-well plates at a density 5 × 10^4^ cells/mL and treated with ferric derisomaltose (100 ng/mL), darbepoetin alfa (10 ng/mL), or ferric derisomaltose plus darbepoetin alfa (100 ng/mL and 10 ng/mL, respectively). Cell death was determined using an annexin V-APC/propidium iodide (PI) apoptosis detection kit (Thermo Fisher Scientific, Waltham, MA, USA). Intact and apoptotic cells were distinguished by flow cytometry (CytoFLEX; Beckman Coulter, Brea, CA, USA), and the respective percentages were calculated using CytExpert (Beckman Coulter).

### 4.5. Erythropoietin Concentration

Concentrations of EPO in cell culture medium and human serum were determined using Human EPO Duoset™ ELISA and Human EPO Quantikine^®^ ELISA kits (R&D Systems, Minneapolis, MN, USA).

### 4.6. RNA-Sequencing Analysis

F36E cells treated with darbepoetin alfa alone or with a combination of ferric derisomaltose plus darbepoetin alfa were subjected to RNA-sequencing (RNA-seq) analysis. Briefly, 100-bp paired-end sequence reads were generated for RNA-seq analysis using the Illumina HiSeq 2500 platform (Illumina, San Diego, CA, USA). The raw reads were stored in FASTQ format, and “dirty” raw reads were removed prior to data analysis. Unique reads mapped to the UCSC hg19 genome assembly were used to calculate gene expression levels, which were measured based on the number of mapped reads. We identified significantly (*p* ≤ 0.01) differentially expressed genes (DEGs) between paired tumors and normal samples. Sequencing was performed by Theragen Bio Institute (Seongnam, Republic of Korea).

### 4.7. Study Subjects and Blood Collection

Selection criteria: (1) Those who signed a written consent form for study participation; (2) Adults aged 19 years or older; (3) Histologically diagnosed advanced/metastatic solid tumor; (4) Patients who received myelosuppressive chemotherapy for palliative purposes within 1 month of study participation and who are scheduled to proceed with chemotherapy during this study; (5) Anemia with functional iron deficiency (hemoglobin < 10 g/dL and functional iron-deficiency status: transferrin saturation < 50% and serum ferritin 30–800 ng/mL); (6) ECOG performance status 0–2; (7) Expected life expectancy ≥ 24 weeks

Exclusion criteria: (1) Absolute iron deficiency (serum ferritin < 30 ng/mL and transferrin saturation < 20%) or no iron deficiency (serum ferritin ≥ 800 ng/mL or transferrin saturation ≥ 50%); (2) Other causes of anemia other than chemotherapy-induced anemia (e.g., vitamin B12 or folic acid deficiency, hemolytic anemia, myelodysplastic syndrome, etc.); (3) Ongoing bleeding at the time of study enrollment; (4) Patients who require rapid blood transfusion at the time of study enrollment according to the investigator’s judgment (e.g., rapidly progressing anemia); (5) Bone marrow invasion by tumor; (6) History of erythropoiesis-stimulating agents within 3 weeks of clinical study enrollment or oral or intravenous iron administration or blood transfusion within 2 weeks of study enrollment; (7) History of venous thromboembolism within 6 months or taking anticoagulants at the time of study enrollment; (8) History or family history of hemochromatosis; (9) Previous hypersensitivity to iron preparations and erythropoiesis-stimulating agents; (10) Uncontrolled acute or chronic infection; (11) Patients with renal dysfunction (serum creatinine ≥ 2.0 mg/dL, or glomerular filtration rate < 30 mL/min/1.73 m^2^) and hepatic dysfunction (AST or ALT ≥ 3 times the upper limit of the normal range); (12) Pregnant or lactating women.

Blood samples were collected before and after treatment with ferric derisomaltose plus darbepoetin alfa (Group I) or darbepoetin alfa alone (Group II) from 27 cancer patients with FIDA who received chemotherapy. Blood samples were obtained from each patient during chemotherapy between November 2022 and October 2024 at Hallym University Medical Center. Blood samples were stored at −80 °C. This study was approved by the ethics committee of Hallym University Medical Center. Written informed consent was obtained from all of the participants. Baseline characteristics of subjects are shown in Table 1.

### 4.8. Statistical Analysis

The data were analyzed by the Mann–Whitney U test using GraphPad Prism 5 (GraphPad Software Inc., La Jolla, CA, USA) and are presented as the mean ± standard error of the mean (SEM). In all analyses, *p* < 0.05 was taken to indicate statistical significance.

## 5. Conclusions

The results of this in vitro study strongly supported the clinical evaluation of ferric derisomaltose plus darbepoetin alfa. In addition, cancer patients with FIDA in the ferric derisomaltose plus darbepoetin alfa combination group showed greater improvement in anemia than those in the darbepoetin alfa monotherapy group. Further large clinical trials are required to expand the scope of these potential new therapeutic options for clinical use.

## Figures and Tables

**Figure 1 ijms-26-02203-f001:**
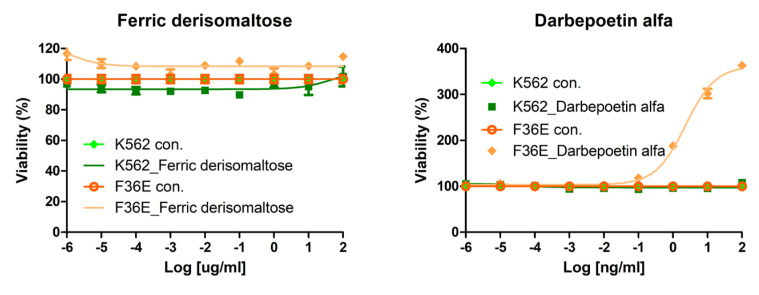
Effects of darbepoetin alfa and ferric derisomaltose on cell viability in acute myeloid and erythroid leukemia cell lines. Two acute myeloid and erythroid leukemia cell lines were treated with various concentrations of darbepoetin alfa or ferric derisomaltose for 48 h.

**Figure 2 ijms-26-02203-f002:**
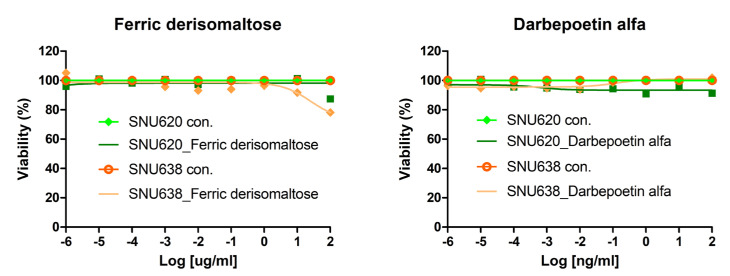
Effects of darbepoetin alfa and ferric derisomaltose on cell viability in GC cell lines. Two GC cell lines, SNU620 and SNU638 cells, were treated with various concentrations of darbepoetin alfa or ferric derisomaltose for 48 h.

**Figure 3 ijms-26-02203-f003:**
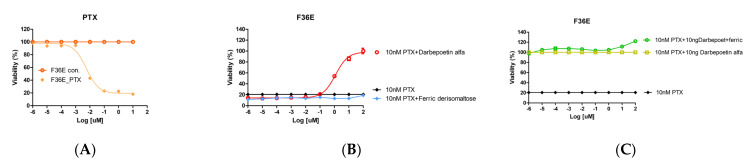
Effects of ferric derisomaltose and darbepoetin alfa on cell viability in erythroid F36E cells. F36E cells were treated with 10 nM of PTX plus various concentrations of ferric derisomaltose and/or darbepoetin alfa for 48 h. (**A**) F36E cells were treated with various concentrations of PTX. (**B**) F36E cells were treated with 10 nM of PTX plus various concentrations of ferric derisomaltose or darbepoetin alfa. (**C**) F36E cells were treated with 10 nM of PTX, 10 nM of PTX plus 10 nM of darbepoetin alfa and 10 nM of PTX plus 10 nM of darbepoetin alfa plus various concentrations of ferric derisomaltose. PTX, paclitaxel.

**Figure 4 ijms-26-02203-f004:**
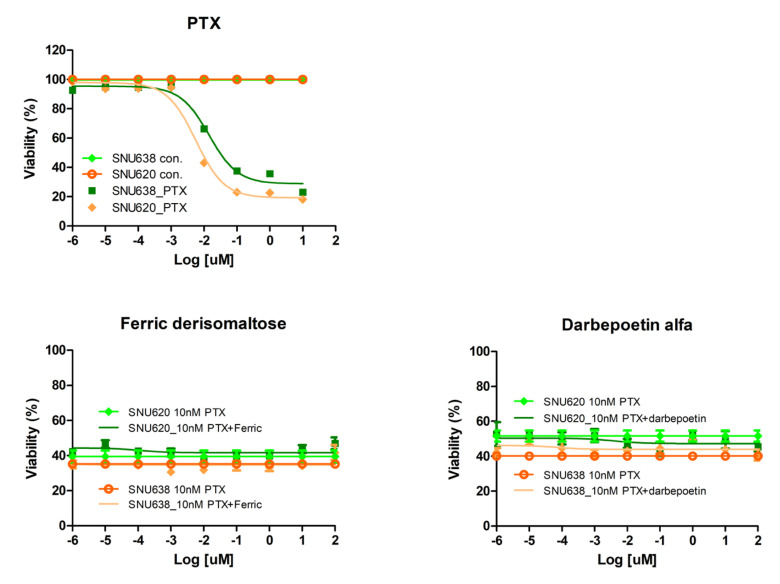
Effects of ferric derisomaltose, darbepoetin alfa, and PTX on cell viability in GC cell lines. SNU620 and SNU638 GC cells were treated with 10 nM PTX plus various concentrations of ferric derisomaltose or darbepoetin alfa for 48 h. PTX, paclitaxel.

**Figure 5 ijms-26-02203-f005:**
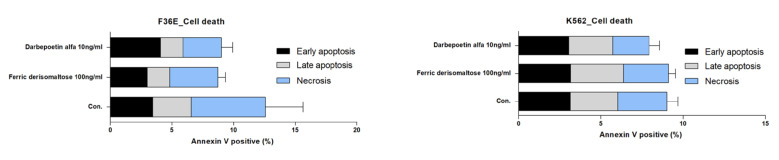
Effects of ferric derisomaltose and darbepoetin alfa on apoptosis and necrosis in myeloid K562 and erythroid F36E cells. The two cell lines were treated with 10 ng/mL of darbepoetin alfa or 100 ng/mL of ferric derisomaltose for 48 h.

**Figure 6 ijms-26-02203-f006:**
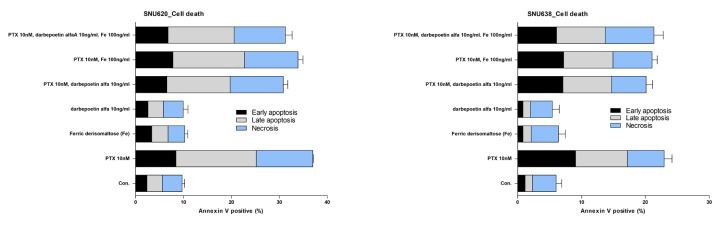
Effects of ferric derisomaltose, darbepoetin alfa, and ferric derisomaltose plus darbepoetin alfa on PTX-induced cell death of GC cell lines. SNU620 and SNU638 cells were treated with 20 nM of PTX, 10 ng/mL darbepoetin alfa (ESA), and/or 100 ng/mL ferric derisomaltose for 48 h. PTX, paclitaxel.

**Figure 7 ijms-26-02203-f007:**
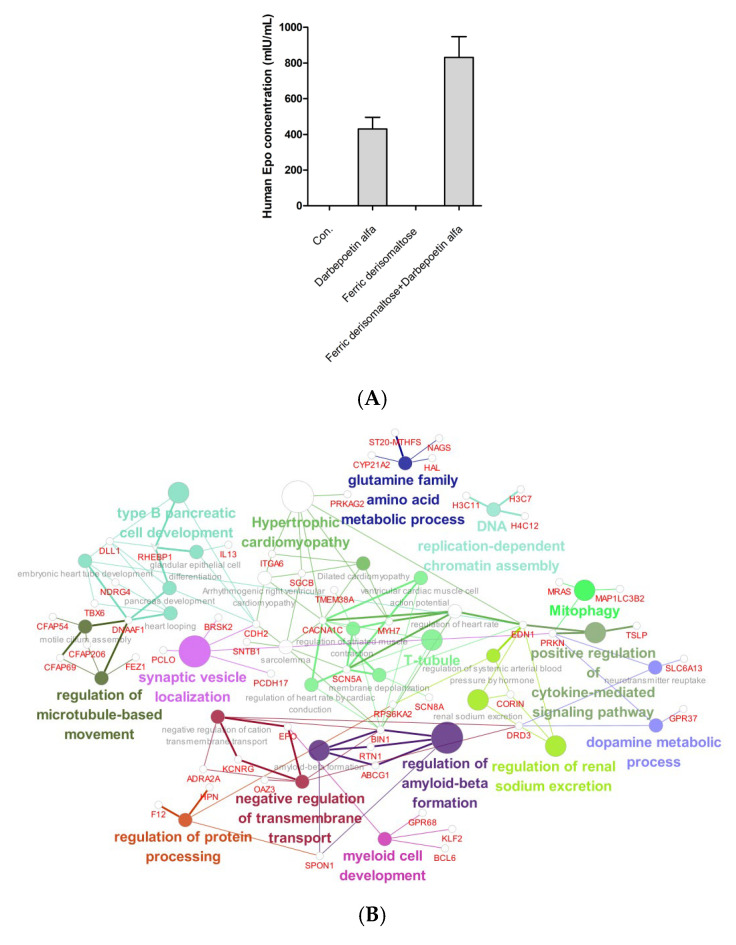
Effects of ferric derisomaltose, darbepoetin alfa, and ferric derisomaltose plus darbepoetin alfa on EPO production and gene expression in F36E cells. F36E cells were treated with 10 ng/mL darbepoetin alfa, 100 ng/mL ferric derisomaltose, or a combination of 10 ng/mL darbepoetin plus 100 ng/mL ferric derisomaltose for 48 h. (**A**) EPO protein was analyzed in cell culture medium by ELISA. F36E erythroid cells were treated with 10 ng/mL darbepoetin alfa vs. 100 ng/mL ferric derisomaltose plus 10 ng/mL darbepoetin alfa for 48 h, and genes that were (**B**) upregulated or (**C**) downregulated were examined by KEGG pathway and gene ontology (GO) analyses.

**Figure 8 ijms-26-02203-f008:**
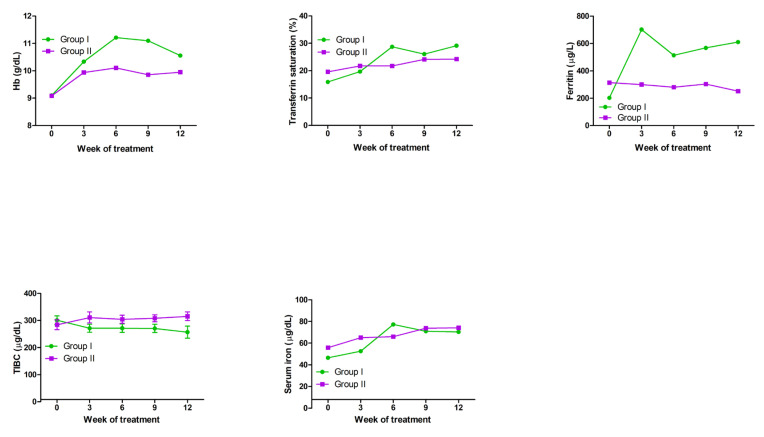
Mean changes in Hb level, transferrin saturation, ferritin level, TIBC, and serum iron level after treatment with intravenous ferric derisomaltose plus darbepoetin alfa (Group I) or darbepoetin alfa (Group II). Hb, hemoglobin; TIBC, total iron-binding capacity.

**Table 1 ijms-26-02203-t001:** Baseline characteristics of 27 subjects.

Group	Subject No.	Sex/Age
Group 1	s-001-001	M/87
	s-001-005	F/70
	s-001-009	M/64
	s001-010	M/67
	s001-012	M/73
	s001-013	M/76
	s001-018	F/55
	s001-020	F/81
	s001-022	M/68
	s001-024	M/58
	s001-026	M/61
	s001-027	F/64
	s001-031	M/53
	s001-032	F/60
Group 2	s-001-008	F/67
	s-001-007	F/68
	s001-011	M/40
	s001-014	F/74
	s001-016	M/57
	s001-017	F/70
	s001-019	M/67
	s001-021	M/66
	s001-025	F/67
	s001-028	M/58
	s001-029	M/62
	s001-030	M/87
	s001-033	M/63

## Data Availability

All the data are presented in the body of the manuscript. The data are available from the corresponding author upon reasonable request.

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
