# Peer review of "Effects of Darbepoetin Alfa and Ferric Derisomaltose Plus Darbepoetin Alfa in Functional Iron-Deficiency Anemia"

_ijms, 2025, doi:10.3390/ijms26052203_

Round 1

Reviewer 1 Report

Comments and Suggestions for Authors

The manuscript is clear, relevant to the field, and presented in a well-structured manner. 

The authors should describe in the introduction the action of Darbepoetin Alfa and Ferric Derisomaltose as such information is missing.
Figures are appropriate and adequately represent data, are easy to interpret and understand.
The cited references are mainly the latest and most important publications, the work does not contain an excessive amount of self-citations.

Accept after Minor Revisions: The paper can in principle be accepted after revision based on the reviewer’s comments. 

Author Response

Point-by-point response to Comments and Suggestions for Authors

Comments 1: The manuscript is clear, relevant to the field, and presented in a well-structured manner.

The authors should describe in the introduction the action of Darbepoetin Alfa and Ferric Derisomaltose as such information is missing.

Figures are appropriate and adequately represent data, are easy to interpret and understand.

The cited references are mainly the latest and most important publications, the work does not contain an excessive amount of self-citations.

Accept after Minor Revisions: The paper can in principle be accepted after revision based on the reviewer’s comments.

Response 1: Thank you for pointing this out. We agree with this comment. We added sentence “Darbepoetin Alfa binds to the EpoR on erythroid progenitor cells, stimulates increasing RBC production and RBC hemoglobinization (erythropoiesis) by the same mechanism as endogenous erythropoietin [11,12]. Ferric derisomaltose (third-generation high-dose intravenous iron supplement) release iron then transported to erythroid precursor cells for incorporation into the hemoglobin [13,14].” in line 57-62.

Reviewer 2 Report

Comments and Suggestions for Authors

I have read the literature entitled "Effects of Darbepoetin Alfa and Ferric Derisomaltose Plus Darbepoetin Alfa in Functional Iron Deficiency. The research is innovative and the research content is also very important. In this study, they investigated the effect of Darbepoetin Alfa combined with Ferric Derisomaltose in the treatment of functional iron deficiency anemia (FIDA) and its mechanism. Overall, this study provides new ideas for the treatment of chemotherapy-induced anemia, especially the study of iron deficiency anemia and dabiprol sodium combined therapy, which is worthy of attention. However, authors need to address the following questions before submitting an article:

I have read the literature entitled "Effects of Darbepoetin Alfa and Ferric Derisomaltose Plus Darbepoetin Alfa in Functional Iron Deficiency. The research is innovative and the research content is also very important. In this study, they investigated the effect of Darbepoetin Alfa combined with Ferric Derisomaltose in the treatment of functional iron deficiency anemia (FIDA) and its mechanism. Overall, this study provides new ideas for the treatment of chemotherapy-induced anemia, especially the study of iron deficiency anemia and dabiprol sodium combined therapy, which is worthy of attention. However, authors need to address the following questions before submitting an article:

  • The article lacks a detailed explanation of the basis for dose selection. Why Darbepoetin Alfa concentrations of 10 ng/mL and 100 ng/mL were selected? Are these choices supported by previous literature? It is recommended to add relevant evidence.
  • For the part of flow cytometry analysis of apoptosis, the detailed description of flow cytometry parameters is lacking, and it is recommended to supplement the detailed operating conditions.
  • Some abbreviations that appear in the text, such as FIDA, should be explained the first time they are used.
  • All charts need to be explicitly referenced in the body. You can point to relevant charts directly when discussing the results.
  • The methods section mentions that 27 patients with FIDA were enrolled in the study, but the screening criteria for patients are not described in detail. Patient inclusion and exclusion criteria, characteristics (such as age, sex, medical history, etc.) should be clearly listed.
  • The article mentions that different concentrations of the drug were used in the experiment, but does not give complete dose-response data.
  • Some terms in this paper are written in different forms, such as Darbepoetin alfa and Darbepoetin Alfa. Note the unity of terms.
  • The structure of some sentences is more complicated, which may lead to difficulty in understanding; At the same time, there may be inconsistent use of tenses in some places.
  • In order to improve the logic and readability of the article, it is recommended to cite the following articles: PMID: 39357438 at page 9 lines 269-272, PMID: 36205374 at page 9 lines 276-278; It is suggested to add an introduction to apoptosis in the Introduction and cite the following literature: PMID: 38160834.

Author Response

Point-by-point response to Comments and Suggestions for Authors

I have read the literature entitled "Effects of Darbepoetin Alfa and Ferric Derisomaltose Plus Darbepoetin Alfa in Functional Iron Deficiency. The research is innovative and the research content is also very important. In this study, they investigated the effect of Darbepoetin Alfa combined with Ferric Derisomaltose in the treatment of functional iron deficiency anemia (FIDA) and its mechanism. Overall, this study provides new ideas for the treatment of chemotherapy-induced anemia, especially the study of iron deficiency anemia and dabiprol sodium combined therapy, which is worthy of attention. However, authors need to address the following questions before submitting an article:

Comments 1: The article lacks a detailed explanation of the basis for dose selection. Why Darbepoetin Alfa concentrations of 10 ng/mL and 100 ng/mL were selected? Are these choices supported by previous literature? It is recommended to add relevant evidence.

Response 1: Thank you for pointing this out. We agree with this comment. Darbepoetin alfa at concentrations of 10 ng/mL or higher resulted in more than 100% increases in viability of F36E erythroid cells treated with 10 nM PTX. Therefore, darbepoetin alfa was selected at a concentration of 10 ng/mL. We added sentence “Therefore, a concentration of 10 ng/mL was selected for darbepoetin alfa.” in line 108-109.

An increase in viability of approximately 20% was observed in cultures when a high concentration of 100 ng/mL ferric derisomaltose was added to 10 nM PTX plus 10 ng/mL darbepoetin alfa. We added sentence “Therefore, a concentration of 100 ng/mL was selected for ferric derisomaltose.” in line 114-115.

Comments 2: For the part of flow cytometry analysis of apoptosis, the detailed description of flow cytometry parameters is lacking, and it is recommended to supplement the detailed operating conditions.

Response 2: Thank you for pointing this out. We agree with this comment. We attached supplement file. We added words “and File S1” in line 141 and 154.

Comments 3: Some abbreviations that appear in the text, such as FIDA, should be explained the first time they are used.

Response 3: Thank you for your comment. Introduction section: we mentioned "Functional iron deficiency anemia (FIDA) is a condition in which the body has sufficient iron stores but cannot use it effectively, which is distinct from iron deficiency anemia where the level of iron in the body is low.” in line 34-37.

Comments 4: All charts need to be explicitly referenced in the body. You can point to relevant charts directly when discussing the results.

Response 4: Thank you for pointing this out. We agree with this comment. We added words “(Figure 8)” in line 244; “(Figure 7)” in line 250; “(Figure 3)” in line 254; “(Figure 5 and 6)” in line 256; “(Figure 4 and 6)” in line 257.

Comments 5: The methods section mentions that 27 patients with FIDA were enrolled in the study, but the screening criteria for patients are not described in detail. Patient inclusion and exclusion criteria, characteristics (such as age, sex, medical history, etc.) should be clearly listed.

Response 5: Thank you for pointing this out. We agree with this comment. We added sentences “Selection criteria: 1) Those who signed a written consent form for study participation; 2) Adults aged 19 years or older; 3) Histologically diagnosed advanced/metastatic solid tumor; 4) Patients who received myelosuppressive chemotherapy for palliative purposes within 1 month of study participation and who are scheduled to proceed with chemotherapy during this study; 5) Anemia with functional iron deficiency (Hemoglobin <10 g/dL and Functional iron deficiency status: transferrin saturation <50% and serum ferritin 30-800 ng/mL); 6) ECOG performance status 0-2; 7) Expected life expectancy ≥ 24 weeks

Exclusion criteria: 1) Absolute iron deficiency (serum ferritin <30 ng/mL and transferrin saturation <20%) or no iron deficiency (serum ferritin ≥800 ng/mL or transferrin saturation ≥50%); 2) Other causes of anemia other than chemotherapy-induced anemia (e.g., vitamin B12 or folic acid deficiency, hemolytic anemia, myelodysplastic syndrome, etc.); 3) Ongoing bleeding at the time of study enrollment; 4) Patients who require rapid blood transfusion at the time of study enrollment according to the investigator's judgment (e.g., rapidly progressing anemia); 5) Bone marrow invasion by tumor; 6) History of erythropoiesis-stimulating agents within 3 weeks of clinical study enrollment or oral or intravenous iron administration or blood transfusion within 2 weeks of study enrollment; 7) History of venous thromboembolism within 6 months or taking anticoagulants at the time of study enrollment; 8) History or family history of hemochromatosis; 9) Previous hypersensitivity to iron preparations and erythropoiesis-stimulating agents; 10) Uncontrolled acute or chronic infection; 11) Patients with renal dysfunction (serum creatinine ≥2.0 mg/dL, or glomerular filtration rate <30 mL/min/1.73m2) and hepatic dysfunction (AST or ALT ≥3 times the upper limit of the normal range); 12) Pregnant or lactating women” in line 305-327.

We added Table 1.

Comments 6: The article mentions that different concentrations of the drug were used in the experiment, but does not give complete dose-response data.

Response 6: Thank you for pointing this out. We agree with this comment. The drug we use is CKD-11101. We added words and reference “(CKD-11101, Nesbell; Chong Kun Dang Pharm, Seoul, Korea)[22]” Lee, J.H et al paper containg dose-response data.

Comments 7: Some terms in this paper are written in different forms, such as Darbepoetin alfa and Darbepoetin Alfa. Note the unity of terms.

Response 7: Thank you for pointing this out. We changed words “Darbepoetin Alfa” à “Darbepoetin alfa”.

Comments 8: The structure of some sentences is more complicated, which may lead to difficulty in understanding; At the same time, there may be inconsistent use of tenses in some places.

Response 8: Thank you for pointing this out. We got my writing corrected by a native speaker.

Comments 9: In order to improve the logic and readability of the article, it is recommended to cite the following articles: PMID: 39357438 at page 9 lines 269-272, PMID: 36205374 at page 9 lines 276-278; It is suggested to add an introduction to apoptosis in the Introduction and cite the following literature: PMID: 38160834.

Response 9: We appreciate the additional comments and are sorry for causing you trouble. This is a study to elucidate the treatment mechanism of chemotherapy-induced anemia. I don't know where to include the apoptosis content.